# Oocyte Ageing in Zebrafish *Danio rerio* (Hamilton, 1822) and Its Consequence on the Viability and Ploidy Anomalies in the Progeny

**DOI:** 10.3390/ani11030912

**Published:** 2021-03-22

**Authors:** Swapnil Gorakh Waghmare, Azadeh Mohagheghi Samarin, Roman Franěk, Martin Pšenička, Tomáš Policar, Otomar Linhart, Azin Mohagheghi Samarin

**Affiliations:** Research Institute of Fish Culture and Hydrobiology, South Bohemian Research Center of Aquaculture and Biodiversity of Hydrocenoses, Faculty of Fisheries and Protection of Waters, University of South Bohemia in Ceske Budejovice, Zátiší 728/II, 389 25 Vodňany, Czech Republic; swaghmare@frov.jcu.cz (S.G.W.); amohagheghi@frov.jcu.cz (A.M.S.); franek@frov.jcu.cz (R.F.); psenicka@frov.jcu.cz (M.P.); policar@frov.jcu.cz (T.P.); linhart@frov.jcu.cz (O.L.)

**Keywords:** egg storage, embryo development, ploidy anomalies, progeny abnormality, zebrafish AB strain

## Abstract

**Simple Summary:**

The maintenance and manipulation of AB strain zebrafish oocytes at 26 °C was found to be possible for 2 h without incurring a marked reduction in fertilization potential. However, the post-ovulatory ageing of oocytes for 6 h resulted in an almost complete loss of egg viability. All larvae derived from the 4- and 6-h aged oocytes were characterized by physical abnormalities. Ageing oocytes for 4 h resulted in the incidence of ploidy anomalies having a four-fold increase. These results make a valuable contribution with respect to the control of experimental reproduction in zebrafish, which is currently accepted as an excellent model animal.

**Abstract:**

Fish egg quality can be markedly influenced by the oocyte age after ovulation. In this study, we examined the duration of oocyte ageing in the zebrafish (*Danio rerio*) and whether prolonged ageing is associated with the incidence of ploidy anomalies in the resulting embryos. Oocytes were incubated in vitro for 6 h post-stripping (HPS) at 26 °C and fertilized at 2-h intervals. Meanwhile, for eggs fertilized immediately after stripping, the fertilization, embryo survival, and hatching rates started at ~80%; these rates decreased to 39%, 24%, and 16%, respectively, for oocytes that had been stored for 4 h (*p* ˂ 0.05), and there was an almost complete loss of egg viability at 6 HPS. Furthermore, almost 90% of the embryos derived from 6-h aged oocytes died prior to hatching, and all larvae originating from 4- and 6-h aged oocytes showed malformations. The proportion of ploidy abnormal embryos was significantly greater at 4 HPS (18.5%) than at either 0 or 2 HPS (4.7% and 8.8%, respectively). The results revealed that zebrafish oocytes retained their fertilization potential for up to 2 h after stripping at 26 °C and indicated the contribution of post-ovulatory oocyte ageing in the occurrence of ploidy anomalies in the resulting embryos.

## 1. Introduction

Oocytes, the gametes of female organisms, are the final products of oogenesis, during which all essential components, such as maternal mRNAs, proteins, lipids, carbohydrates, vitamins, and hormones, that support initial embryonic development are incorporated [1]. These maternally incorporated endogenous factors are important for ensuring the production of good-quality eggs, and they subsequently contribute to successful fertilization and successful embryo development [2,3]. Oocytes undergo an ageing process with a prolongation in the time span between ovulation and fertilization that is known as post-ovulatory oocyte ageing, during which there is an accumulation of changes in the maternally derived egg components and a concomitant reduction in egg quality. Oocyte ageing appears to be a complex process involving multiple pathways. Some of the cellular and molecular changes associated with this ageing process have been investigated in fish [4,5,6], as well as in other vertebrates [7,8]. However, the precise pathways underlying oocyte ageing have yet to be sufficiently characterized.

Limited fertilization capacity [9,10,11] and disturbed embryonic development [12,13] are among the major phenotypical and functional consequences of oocyte ageing. Prolonged or retracted development during embryogenesis is frequently observed in embryos arising from more aged oocytes. Moreover, although oocytes that undergo relatively shorter periods of post-ovulatory ageing can still produce eyed and hatched embryos, these can be prone to post-hatching abnormalities. For example, fish larvae that develop from aged oocytes are often characterized by corporal malformations [9,12,14], and ploidy anomalies in progeny have been reported among the detrimental outcomes of oocyte ageing, with an increased occurrence of ploidy anomalies being observed in trout (*Oncorhynchus mykiss*) [14], tench (*Tinca tinca*) [15], Japanese eel (*Anguilla japonica*) [16], northern pike (*Esox lucius*) [17], and yellowtail tetra (*Astyanax altiparanae*) [13]. However, the consequence of oocyte ageing on ploidy abnormalities has not yet been studied in zebrafish despite it being widely used as a model organism. Oocyte ageing may also have long-lasting repercussions on progeny, leading to a heightened susceptibility to certain disorders later in life [18].

Zebrafish are valued for their ornamental characteristics, but they mainly serve as an excellent model animal basic biology research [19,20,21]. Given their small size, the housing and care of zebrafish are relatively simple and inexpensive. Additionally, unlike many other fish species, zebrafish mature rapidly within two to four months, and they can also reproduce throughout the year. The short life cycle of this tiny fish thus makes it eminently suitable for multi-generational studies [22]. Moreover, zebrafish eggs, embryos, and larvae are transparent, which is conducive for experimental observations. In comparison with other model organisms, zygotic genome activation is delayed in zebrafish, thereby enabling research, whereas embryonic development is still controlled by maternal components [23]. Furthermore, the feasibility of zebrafish egg storage enables the provision of synchronized female and male gametes in artificial reproduction schedules, which also has advantages with respect to the control of experimental reproduction and the manipulation of oocytes in associated studies. Capitalizing on the aforementioned advantages, we used zebrafish in the present study to investigate the effect of post-ovulatory oocyte ageing on egg fertilization potential, embryo development, larval malformations, and the incidence of ploidy anomalies in progeny.

## 2. Materials and Methods

The effect of post-postovulatory oocyte ageing on the viability and ploidy anomalies in the zebrafish (AB strain) progeny was studied. Following ovulation, oocytes obtained from 7 females were stored separately in vitro for 6 h and artificially inseminated with 2 h-intervals.

### 2.1. Broodfish

Generations from zebrafish (AB strain) that were initially obtained from the European Zebrafish Resource Center (Karlsruhe, Germany) were used as the broodstock. The fish were kept in the rearing system (ZebTec Active Blue). The fish were cultured under controlled conditions (14-h light:10-h dark photoperiod; temperature: 28 °C; feeding: TetraMin flakes twice daily and Artemia nauplii once daily). The experimental fish were aged approximately 12 months with an average weight of 0.57 g (males) and 0.81 g females). During the afternoon of the day prior to commencing experiments, 10 pairs of broodfish were separately placed (one male and one female) into 1-L spawning chambers, within which the males and females were initially separated using a barrier. At light onset on the following day, the barriers were removed, and the fish were thereafter inspected for ovulation. Upon observation of a few eggs at the bottom of a chamber, the females (7) were assumed to have ovulated. The experimental fish were immediately anaesthetized using 0.05% tricaine methanesulfonate (methyl-aminobenzoate, MS222) for the purpose of gamete collection [24].

### 2.2. In Vitro Oocyte Ageing and Sampling

Ovulated oocytes were collected from each female by hand stripping and separately placed into sterile Nunc glass-bottomed dishes (40 mm in diameter) (Thermo Fisher Scientific, Rochester, NY, USA), without the addition of any artificial media. A small piece of moistened tissue was placed adjacent to the eggs to maintain a sufficient level of humidity. Having replaced the lids, the dishes were stored in a laboratory incubator (Q-cell, 140/40 Basic; Wilkowice, Poland) at 26 °C for 6 h. Stored oocytes were fertilized at 0 (immediately after stripping), 2, 4, and 6 h post-stripping (HPS).

### 2.3. Artificial Insemination

Prior to artificial fertilization, the stripped milt from each male was examined for sperm motility under a Nikon SMZ745T stereomicroscope (Nikon, Japan) according to Fauvel et al. [25]. Given that zebrafish males generally produce small quantities of milt (typically not exceeding 2 µL) and to provide a uniform fertilization potential for all egg batches, sperm was collected from five males and pooled to give a composite sample for the purposes of artificial fertilization at each time point. The pooled milt was added to a tube containing 50 µL of an immobilizing solution (Kurokura 180 solution) [26]. For the artificial fertilization at each HPS, batches of ~100 oocytes were placed in Petri dishes, to which 10 μL of the pooled sperm followed by 0.2 mL of water were added. Having been shaken for 1 min, egg batches were washed by immersing in 3 mL of water. Preliminary evaluations enabled us to confirm that the oocyte:sperm:water ratio used in the present study was sufficient to ensure the fertilization of all eggs. The eggs thus inseminated were thereafter maintained in Petri dishes (9 cm in diameter) at 28 °C in a laboratory incubator.

### 2.4. Examination of Egg Developmental Success and Larval Quality

As egg quality indices, we examined the rates of fertilization, embryo survival, hatching, embryo mortality, and larval malformation for eggs originating from oocytes of different ages. The calculations were made as below:(1)Fertilization%: (number of eggs showing cleavage/total number of inseminated eggs) × 100.(2)Embryo survival%: (number of 24-h live embryos/total number of inseminated eggs) × 100.(3)Hatching%: (number of hatched larvae/total number of inseminated eggs) × 100.(4)Egg mortality%: (number of dead embryos/total number of 24-h survived embryos) × 100.(5)Larval malformation%: (number of morphologically abnormal larvae/total number of hatched larvae) × 100.

Fertilization rates were assessed for each egg batch 90 min after egg insemination, based on the occurrence of cell divisions observed under a Nikon SMZ745T stereomicroscope. Egg batches were re-examined at 24 h after insemination to evaluate embryo survival. The embryo mortality was measured between 24 h after the insemination and the hatching. After hatching (48–72 h after insemination), larvae were observed under a stereomicroscope in order to determine the occurrence of any morphological abnormalities (e.g., shortened body, deformed tail, spinal cord torsion, deformed yolk sac, and eye deformations) [27], thereby facilitating an estimate of the extent of larval malformation.

### 2.5. Examination of Ploidy Levels in Progeny

The potentially adverse effects of post-ovulatory oocyte ageing on the ploidy level of progeny emerging from eggs developed from oocytes of different ages were assessed on the first day post-fertilization and again immediately after hatching, based on flow cytometric measurements of the relative DNA content of nuclei. Detailed information relating to the number of analyzed embryos and larvae derived from oocytes at each HPS is presented in Table 1. The experimental embryos and larvae were processed using a CyStain UV Precise T kit for nuclei staining (Sysmex Partec GmbH, Münster, Germany) according to the manufacturer’s protocol, with the following modifications. Single embryo/larva were homogenized in 200 μL of the extraction buffer. Samples were then filtered through 30-μm mesh filters (Partec Cell Trics disposable filter units; Partec). Subsequently, 1 mL of the staining buffer was added to the nuclei suspension and analyzed by flow cytometry at a flow rate of 0.4 µL s^−1^. The relative DNA content was determined using a CyFlow Ploidy Analyser (Sysmex Partec GmbH) against samples from the diploid control group (eggs fertilized at 0 HPS).

### 2.6. Statistical Analysis

Data analysis was performed using the R (Version 4.0.3) statistical program. The normality of the data was assessed using histogram. Following ANOVA, Duncan’s multiple range test was used to analyze the data, and differences were considered significant at the *p* < 0.05 level.

## 3. Results

### 3.1. Egg Quality Indices of Ageing Oocytes

The highest fertilization, embryo survival, and hatching rates (~80%) were observed for those eggs fertilized immediately after stripping (Figure 1). Though not significantly different, we detected a reduction in the rate of fertilization to 64% for the eggs derived from 2-h aged oocytes. Thereafter, the fertilization rates significantly decreased and dropped to only 3% at 6 HPS. The percentages of embryo survival were, however, significantly impaired, decreasing to 51%, 24%, and 2% at 2, 4, and 6 HPSs, respectively. The hatching rates followed the same trend of significant decrease through oocyte ageing, and only less than 1% of the eggs hatched at 6 HPS. Moreover, we observed significant increases in the rates of embryo mortality and larval malformation after 4 HPS, with 73% and 90% of the fertilized eggs originating from the oocytes aged for 4 and 6 HPS, respectively, dying prior to hatching (Figure 2). Meanwhile, only 5% of the control group (0 HPS) larvae showed any evidence of malformation; all the larvae originating from 4- and 6-h aged oocytes were observed to be malformed, mostly suffering from the skeletal abnormalities.

### 3.2. Ploidy Anomalies of Ageing Oocytes

One-day-old embryos derived from oocytes aged for 0 and 2 h prior to insemination were characterized by 4.7% and 8.8% ploidy anomalies, respectively (Table 1). Ageing oocytes for 4 h was observed to result in a continuous increase in the percentage of anomalies inone1-day-old embryos, with a four-fold increase (18.5%) in the incidence of ploidy anomalies being detected, as assessed by flow cytometry. A majority of these ploidy anomalies were manifested as triploids (40%) and haploids (40%), whereas the remaining 20% of anomalies were mosaics (n/3n).

In contrast, however, post-ovulatory oocyte ageing appeared to have no appreciable influence on the ploidy status of larvae, with the examination of the larvae (freshly hatched) obtained from seven females at 0, 2, and 4 HPS revealing that all were normal diploids. Due to the very limited number of alive embryos and larvae at 6 HPS, the ploidy levels were not possible to be examined at this time point.

## 4. Discussion

In fish, an inability to achieve optimal timing for fertilization might occur due to a delay in egg spawning or stripping. This, in turn, would result in oocyte ageing, which has been identified as the main factor that impedes successful fertilization and normal embryonic and larval development. The optimal time for egg fertilization following ovulation is typically species-specific and dependent on individual female and storage temperature [28]. In the current study, we detected no significant differences in the fertilization potential of zebrafish (AB strain) oocytes aged in vitro at 26 °C for up to 2 h after stripping, although we did note a decreasing trend during this period. A number of previous studies have examined the effects of storage conditions on oocyte viability. Sakai et al. [29], for example, assessed the effects of storing wild strain zebrafish oocytes at 23 °C in Hanks’ buffered saline supplemented with 0.5% bovine serum albumin (BSA); they found that the stored oocytes retained a fertilization capacity of up to 85% within 1 h, and after 2 h of storage, the rate of fertilization rate did not significantly differ from that of eggs stored for 1 h. Similarly, Cardona-Costa et al. [30] used a medium containing Hanks’ saline supplemented with 1.5 g of BSA and 0.1 g of sodium chloride for the storage of golden zebrafish oocytes at 8 °C; based on a monitoring of fertilization capacity and embryonic development, they reported rapid oocyte ageing after less than 1 h. The authors accordingly recommend that fertilization should be conducted as soon as possible after collecting eggs from females and noted that the strain of experimental zebrafish should be taken into consideration when seeking to determine the optimal period for oocyte fertilization.

There is currently little available information regarding the cellular and molecular processes associated with the lower fertilization rates observed for aged oocytes compared with fresh oocytes. It has, however, been indicated that during the fertilization of aged oocytes, there is an impairment of cytoplasmic Ca^2+^ homeostasis, which is followed by abnormal oscillations in Ca^2+^ that are characterized by a higher frequency but a lower amplitude than those of fresh oocytes [31,32,33]. Similar oscillations have also been reported as a probable cause of the apoptosis pathway associated with abnormal fertilization [34]. Further factors that could potentially account for the reduced fertilization rate of aged oocytes are an increase in oxidative stress and elevated levels of lipid peroxidation in the oocyte plasma membrane [32]. In response to an increase in lipid peroxidation, there is a reduction in membrane fluidity, which can affect the fusion between spermatozoa and the oolemma, thereby reducing the rate of fertilization [35]. However, although these findings provide a superficial insight into the factors contributing to the poor fertilization capacity of aged oocytes, the underlying mechanisms are likely to be more complex and thus warrant more in-depth studies in the future.

The developmental competence of fish embryos is critically dependent on the integrity of oocytes, and our examination of the effects of zebrafish oocyte ageing on embryo survival and hatching rates in the present study revealed that the corresponding rates were significantly lower in the eggs derived from oocytes aged for 2, 4, and 6 h than those derived from fertilized fresh oocytes. These observations accordingly indicated the pivotal importance of oocyte ageing in the deterioration of egg and embryo quality. In this regard, we observed that oocytes subjected to a 6-h post-ovulatory ageing were characterized by defective cleavages. Indeed, only a few embryos (2%) progressed to the gastrula stage, and further development was rarely observed. In contrast to the findings of the present study, Cardona-Costa et al. [30] found that after only 1 h of oocyte ageing at 8 °C in zebrafish (golden strain), the development of embryos to the mid-blastula stage was reduced to 50% compared with the embryos derived from fresh oocytes. Moreover, these authors reported an embryo survival of only 1.8% at 2 HPS and no embryonic development at 3 and 4 HPS. In the present study, we observed that more than 50% of the embryos derived from 2-h aged oocytes developed to the day after fertilization, which was considerably higher than previously published values and can probably be attributed to differences in storage media, storage temperature, and fish strains. In addition, given that the quality of fish oocytes is likely to be affected by other factors in addition to oocyte ageing, the observed inter-female variability in viability is perhaps not surprising. Previous studies have reported significantly lower hatching rates with an increase in the time elapsed between ovulation and fertilization for oocytes aged both in vivo and in vitro. For example, in species such as rainbow trout [36], Japanese eel [16], northern pike [17], yellowtail tetra [13], and common carp (*Cyprinus carpio*) [37], significant reductions in hatching potential have been observed for eggs with an extension in the time between ovulation and fertilization. In this regard, it has been demonstrated that for a majority of those fish species that have been assessed, a reduction in storage temperature from the natural spawning one results in an extension of the successful egg storage time [12,14,38,39]. Accordingly, in future studies, it would be instructive to examine the effects of storage temperature on the in vitro viability of zebrafish oocytes.

Despite the low fertilization capacity of advanced-aged oocytes, they can still give rise to developed embryos and even hatched larvae. However, such embryos and larvae are typically characterized by prolonged or diminished development during the subsequent stages of embryogenesis and are notably prone to premature mortality and post-hatching aberrations. The extrusion of essential components and proteins of yolk from the oocytes to the ovarian fluid during the ageing period [40] and changes in oocyte biochemistry [10,41] may be among the factors contributing to the poor development of embryos derived from aged oocytes. For example, it is believed that cytoplasmic ageing can substantially hamper normal embryo development [35], as changes in oocyte components at the mRNA and protein levels are detectable with increasing time following ovulation [37,42,43]. In zebrafish, the zygotic genome is activated at the 10th cell cycle division, prior to which embryonic development is controlled by maternal mRNAs and encoded products [2]. Maternal effect genes encode proteins that are determined by the maternal genotype, and their mRNAs influence the phenotypes of offspring [44]. The degradation of many maternal mRNAs that are essential for the proper embryo development is not properly done in aged mouse oocytes [45]. The authors suggested that abnormal patterns of maternal mRNA degradation in aged oocytes can contribute to a reduction in the developmental competence of the resulting embryos, and, consequently, a compromised regulation of maternal genes during post-ovulatory oocyte ageing could affect the developmental potential of the generated embryos.

The results obtained in the present study revealing increased malformation and ploidy anomalies in progeny derived from the more aged oocytes were consistent with the findings of previous studies on other fish species. For example, in species such as the African catfish (*Heterobranchus longifilis*) [46], European catfish (*Silurus glanis*) [9,47], Asian catfish (*Pangasius hypophthalmus*) [48], rainbow trout [14,49], and common carp [37], embryos arising from aged oocytes have been observed to suffer from physical malformations and irregularities. Oocyte ageing in rainbow trout [14], tench [15], Japanese eel [16], northern pike [17], and yellowtail tetra [13] has also been shown to be associated with a heightened occurrence of triploidization. In the present study, we detected no significant differences between the ploidy levels of hatched larvae originating from fresh and aged oocytes, with all hatched larvae being normal diploids. In this regard, although it is generally believed that the viability of triploids is similar to that of diploids in many species of fish [50], the results of the current study indicated that ploidy abnormal embryos derived from aged oocytes are less likely to survive than are diploids. This was previously shown for the European catfish as well [51]. This was apparent from our finding based on flow cytometric analysis, which revealed that the percentage of abnormal ploidy individual decreased as embryos developed to larvae, with no ploidy anomalies being detected among those larvae that hatched. Indeed, all hatched larvae were found to be normal diploids. These observations are consistent with the findings of our study on common carp, which indicated that ploidy abnormal embryos derived from post-ovulatory aged oocytes do not reach the hatching stage (Samarin et al., unpublished data). Thus, it is plausible that it is only triploids that develop as a consequence of post-ovulatory oocyte ageing that are less likely to survive, given that the spontaneous occurrence of triploid larvae can be attributed to several factors [16], among which post-ovulatory oocyte ageing is only one of them. Therefore, the ploidy abnormalities caused by the ageing of oocytes are probably a suitable marker for examining the consequence of oocyte ageing on the arising embryos that are already incompatible with life.

Post-ovulatory aged oocytes exhibit chromosomal abnormalities and misalignment due to meiotic spindle defects that could potentially contribute to higher rates of non-viable embryos post-fertilization [35,52]. These aberrations can increase the risk for chromosome separation defects and aneuploidy [53]. Nomura et al. [16] suggested that the occurrence of cytogenetically atypical progeny in Japanese eels is associated with post-ovulatory oocyte ageing, although this appears to be unrelated to sperm ageing. Using genotypic segregation analysis, they found that triploid larvae are derived from duplication of the maternal alleles but not paternal ones. These events can be ascribed to the inhibition of second polar body and subsequent fertilization with sperm. Mitochondrial dysfunction and a reduction in ATP levels are important contributors to chromosomal abnormalities and the poor quality of aged oocytes and embryos [54]. Hamatani et al. [43] demonstrated that a reduction in the function of ATP-dependent proteins, such as those comprising microtubules and the cytoskeleton, could result in chromosomal segregation defects and an associated increase in aneuploidy in aged oocytes. Furthermore, Pan et al. [45] demonstrated that the strength of the spindle assembly checkpoint is weakened with an increase in reproductive age in mouse oocytes, with the authors concluding that a higher proportion of errors in microtubule–kinetochore interactions could be considered the underlying basis for an increase in the extent of aneuploidy during maternal ageing. Accordingly, in future studies, it would be of considerable interest to examine molecular pathways associated with the incidence of ploidy anomalies caused by fish oocyte ageing.

## 5. Conclusions

Post-ovulatory oocyte ageing in zebrafish has a considerable influence on the developmental competence of the resulting eggs. In this study, we found that fertilization rates, embryo survival, and the capacity to reach the hatching stage were markedly reduced with a prolongation of post-ovulatory oocyte ageing. An extension in the time of oocyte ageing of up to 2 h in the maintenance and manipulation of AB strain zebrafish oocytes at 26 °C is, nevertheless, possible without incurring a marked reduction in fertilization potential. In contrast, however, the post-ovulatory ageing of oocytes for 6 h resulted in an almost complete loss of egg viability. The mortality rates of the developing embryos and malformations in the resulting larvae were also found to be adversely affected by post-ovulatory oocyte ageing, with all larvae derived from the 4- and 6-h aged oocytes being characterized by physical abnormalities. We believe that the results obtained in the present study will make a valuable contribution with respect to the control of experimental reproduction in zebrafish, which is currently accepted as an excellent model animal.

## Figures and Tables

**Figure 1 animals-11-00912-f001:**
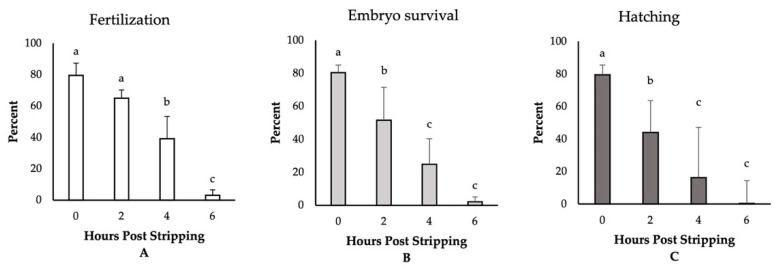
The effect of post-stripping oocyte ageing on the rates of (**A**) fertilization, (**B**) embryo survival, and (**C**) hatching in zebrafish (mean ± SD). Mean values indicated by the same letter (a, b, c) do not differ significantly.

**Figure 2 animals-11-00912-f002:**
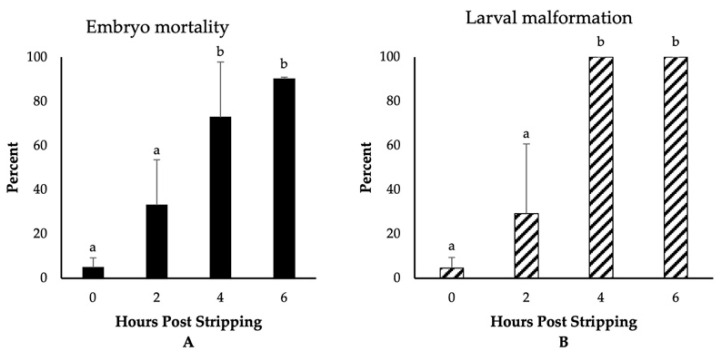
The effect of post stripping oocyte ageing on the rates of (**A**) embryo mortality and (**B**) larval malformation in zebrafish (mean ± SD). Mean values indicated by the same letter (a, b) do not differ significantly.

**Table 1 animals-11-00912-t001:** Effect of post-ovulatory oocyte ageing on ploidy anomalies in the embryos (1-day old) and larvae (immediately post-hatching) of zebrafish.

1 Day Post-Fertilization		
HPS	No. of Analyzed Embryos	Percentage Ploidy Anomalies
0	64	4.7 (100% Tr)
2	34	8.8 (33% H, 33% Te, 33% M)
4	27	18.5 (40% H, 40% Tr, 20% M)
**Larvae Immediately Post-Hatching**		
**HPS**	**No. of Analyzed Larvae**	**Percentage Ploidy Anomalies**
0	40	0
2	65	0
4	20	0

HPS: hours post-stripping; H: haploid; Tr: triploid; Te: tetraploid; M: mosaic.

## Data Availability

Data from the analysis are available from the corresponding author on request.

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
