# Peer review of "Oocyte Ageing in Zebrafish Danio rerio (Hamilton, 1822) and Its Consequence on the Viability and Ploidy Anomalies in the Progeny"

_animals, 2021, doi:10.3390/ani11030912_

Round 1

Reviewer 1 Report

The experiment design is poor.  There is no way to judge what is/are the number of replicates.  Lumping eggs and sperm from a bunch of fish is not appropriate for statistical analysis.  Using individual eggs as individual replicates is "illegal" because they could have come from the same female...and male.  There is no way to untangle these data.  I cannot decipher the 1st figure. What do the bars represent?

The whole study seems trivial.  When do eggs decompose enough so they no longer can become normal fish?  There is no theoretical or scientific foundation to this study.  It certainly has no meaning to the biology of the fish.

Reviewer 2 Report

The manuscript entitled "Aging of oocytes in zebrafish Danio rerio (Hamilton, 1822) and its consequences on abnormalities of vitality and ploidy in progeny", described in a clear and detailed way the effect of post-ovulatory aging of oocytes on the potential for egg fertilization, embryonic development, larval malformations and the incidence of ploidy anomalies in the progeny.

The Introduction section describes the context appropriately and contains the information necessary for the general framework of the study.

The methods used are clearly and precisely defined in the materials and methods, reporting recent bibliographic references in this area.

The results are presented in a logical sequence in the text and are easy to understand thanks also to the data reported in the figures and tables.

The discussions are extensive, well described and also refer to possible future experiments in this area.

Reviewer 3 Report

The present study describes the effects of oocyte aging on zebrafish reproduction rates. I have read this manuscript with great interest since the importance zebrafish holds as an established model organism. In this scenario is essential to deepen our knowledge of reproductive dynamics. The approach is scientifically sound and the manuscript is well written. However, there are several issues, which need further clarification by the authors:

Major comments:

  • In 3.1 I found results particularly difficult to read, in contrast to the simplicity of the outcome. Modulation of the paragraph is recommended. Maybe considering each result separately and on a clearer temporal scale.
  • in my opinion, in Figure 1 the use of letters to identify groups with no significant differences is misleading. Please consider a modulation of the graphs, maybe also splitting it into 3 different ones.

Minor comments:

  • In Paragraph 2.4 a resume table could be useful to explain in a clearer way the calculations for the rate of fertilization, embryo survival, and hatched larvae.
  • In Figure 1. Seems that embryo survival is slightly higher than fertilized eggs how could it be possible?

Reviewer 4 Report

Dear Authors,

your manuscript focus on a very important topic for the aquatic research, based on a highly used experimental model as zebrafish. I found it fluent and really interesting in its content and results. I like really liked your style and your care in discussing the results. There are some self-citations between references, but are justified by your noticeable experience in the field. However, There are some minor revision to address before publication, that I summarize as follow:

Abstract: the section is well written and significative, but however lightly exceed the limit of 200 words (11 words). Please take care of this, if for the editorial process is a problem.

Keywords: please take care to avoid the use between the keywords, of words already reported in title, in order to give more resonance at your manuscript during the web search phases. Try to substitute Danio rerio, Oocyte age and Ploidy, with different others.

Introduction: this section gives a god overview of the field and the treated topic, but there are few references cited in my opinion, considering the amount of stuidies performed on this experimental model and more in general for this topic. Paricularly from line 62 to line 74, you should be support your statement with more citations.

Please see and cite for example:

Salvaggio A, Marino F, Albano M, Pecoraro R, Camiolo G, Tibullo D, Bramanti V, Lombardo BM, Saccone S, Mazzei V, Brundo MV. Toxic Effects of Zinc Chloride on the Bone Development in Danio rerio (Hamilton, 1822). Front Physiol. 2016 Apr 29;7:153. doi: 10.3389/fphys.2016.00153. PMID: 27199768; PMCID: PMC4850361.

Materials and methods: a brief introduction of the section, before starting with 2.1 paragraph, would be inserted to give it more readability. 

In could be interesting for the readers to know the numbers of broodfish that you used in your study, in order to attest from how many specimens you collected you quantity of gametes. And of course, being animal experimentation, declare if you had problems like mortality related to the stripping phases.

Your results are really newsworthy and really well discussed from all the point of view, will certainly be useful for future studies in the fileld, as already highlighted by you in some points of the discussion section. Really a good job.

References: please take care to use italics style for all the species name.

Best regards

The reviewer

Round 2

Reviewer 1 Report

"I admit that I do not understand the significance of zebrafish eggs survival under the conditions you describe.  But, the real problem is that your statistical analysis is still a problem.  If you are going to cite a statistical analysis, such as the ANOVA, you must give the details of the analysis.  For example, when citing a p value, you must give the F value, N, and the degrees of freedom. 

Also, I still do not understand your sample size (N).  You state that you used 7 females and you kept the eggs from each female separate.  I have the impression from your paper that you then combined all of the eggs into one group and then treated each egg as a separate sample.  If that is correct, the N is not 7 or any other number and, in fact, the N = 1.  If this is true, you cannot perform any statistical analysis.  However, if you know the identity of each egg and its survival time, you can generate an average for each female and use those 7 averages in the ANOVA."  

The authors cite using "R"  in analyzing the statistical program "ANOVA" but have absolutely no idea as to how to use it.  I find it very odd that the authors seem unable to clarify the most basic concept such as the sample size.